# Current Classification and Diversity of *Fusarium* Species Complex, the Causal Pathogen of *Fusarium* Wilt Disease of Banana in Malaysia

Anysia Hedy Ujat [1,2], Ganesan Vadamalai [1], Yukako Hattori [2], Chiharu Nakashima [2], Clement Kiing Fook Wong [3] and Dzarifah Zulperi [1,4,*]

1 Department of Plant Protection, Faculty of Agriculture, Universiti Putra Malaysia, Serdang 43400, Selangor, Malaysia; adelia.anysia.hedy@gmail.com (A.H.U.); ganesanv@upm.edu.my (G.V.)
2 Graduate School of Bioresources, Mie University, Tsu, Mie 514-0001, Japan; yuka.pon.qta@gmail.com (Y.H.); chiharu@bio.mie-u.ac.jp (C.N.)
3 Department of Agricultural and Food Science, Faculty of Science, Universiti Tunku Abdul Rahman, Kampar 31900, Perak, Malaysia; kfwong@utar.edu.my
4 Laboratory of Sustainable Resources Management, Institute of Tropical Forestry and Forest Products, Universiti Putra Malaysia, Serdang 43400, Selangor, Malaysia
* Correspondence: dzarifah@upm.edu.my

**Abstract:** The re-emergence of the *Fusarium* wilt caused by *Fusarium odoratissimum* (*F. odoratissimum*) causes global banana production loss. Thirty-eight isolates of *Fusarium* species (*Fusarium* spp.) were examined for morphological characteristics on different media, showing the typical *Fusarium* spp. The phylogenetic trees of *Fusarium* isolates were generated using the sequences of histone gene (*H3*) and translation elongation factor gene (*TEF-1α*). Specific primers were used to confirm the presence of *F. odoratissimum*. The phylogenetic trees showed the rich diversity of the genus *Fusarium* related to *Fusarium* wilt, which consists of *F. odoratissimum*, *Fusarium grosmichelii*, *Fusarium sacchari*, and an unknown species of the *Fusarium oxysporum* species complex. By using Foc-TR4 specific primers, 27 isolates were confirmed as *F. odoratissimum*. A pathogenicity test was conducted for 30 days on five different local cultivars including, *Musa acuminata* (AAA, AA) and *Musa paradisiaca* (AAB, ABB). Although foliar symptoms showed different severity of those disease progression, vascular symptoms of the inoculated plantlet showed that infection was uniformly severe. Therefore, it can be concluded that the *Fusarium oxysporum* species complex related to *Fusarium* wilt of banana in Malaysia is rich in diversity, and *F. odoratissimum* has pathogenicity to local banana cultivars in Malaysia regardless of the genotype of the banana plants.

**Keywords:** *Fusarium oxysporum*; *Fusarium odoratissimum*; banana; molecular characterization

## 1. Introduction

Edible banana, genus *Musa*, belongs to the family *Musaceae* of the *Zingiberales*, is noted for its nutritional content, is rich in potassium, and is one of Asia's staple foods [1,2]. In 2019, the export of bananas in Asia was 4.5 million tonnes and increased 18% compared to 2018 [3]. Banana plants are known as one of the earliest crops to be domesticated [4]. The early banana export trade was established based on Panama's cultivar 'Gros-Michel (AAA)'. However, the fungal wilt disease, caused by *Fusarium oxysporum* f. sp. *cubense* (Foc), was then infested the banana plantations. A while later, the disease spread throughout the cropping areas globally, and the banana plantation suffered catastrophic damage [1]. The cultivar 'Cavendish' resistant to the *Fusarium wilt* was introduced to the farms and replaced 'Gros-Michel (AAA)' in the world. In the 1990s, a different race of the same fungus, Tropical Race 4 (TR4), arose in Southeast Asia. This race can infect and disease the cultivar 'Cavendish', shows similar wilting symptoms, and eventually dies out [5].

About 80% of cultivars of world banana production are susceptible varieties to *Fusarium* wilt, such as 'Cavendish', 'Highland & ABB bananas', and 'Gros-Michel'. However, the susceptibility of various local banana varieties in Malaysia against *Fusarium* wilt caused by *Fusarium oxysporum* f. sp. *cubense* TR4 has not been evaluated yet.

Maryani et al. [6] recently reclassified the taxonomical position of *Fusarium oxysporum* f. sp. *cubense* based on its phylogenetic position and phenotypic characters, such as morphology and pathogenicity. As a result, it has been named *Fusarium odoratissimum*, which is an independent species of the *Fusarium oxysporum* species complex. To date, there is no effective control method against this disease due to the nature of the causal agent itself [7]. Therefore, existing disease management policies, including quarantine procedures and standard operating procedures on agronomic practices [8], need to understand the fungal biodiversity to develop control strategies.

In Malaysia, several studies of the genetic diversity of Foc have been conducted based on the molecular phylogeny and VCG analyses, and those results suggest the existence of highly divergent variations of strains of Foc-TR4 (*F. odoratissimum*) [9,10]. Morphological characterization is a fundamental initial classification that has been used to identify the fungal at the species level [11]. The histone (*H3*) gene was used to classify filamentous ascomycetes and deuteromycetes as it could amplify the conserved gene's introns [12] to observe the polymorphism. Since the development of this primer set, it has been widely used to detect *Fusarium* spp., including *Fusarium oxysporum* and *Fusarium subglutinans* [13–15]. In contrast, the translation elongation factor 1–alpha (*TEF-1α*) has been one of the most reliable species-level markers of choice as it is highly informative for *Fusarium* spp. It was first developed to investigate the *Fusarium oxysporum* species complex lineage [16]. A recent study by Zeng et al. [14] shows that both the *H3* gene and *TEF-1α* are comparable and produced powerful resolution.

This study aims to grasp the current diversity of the *Fusarium* species complex infecting the Malaysian local banana related to the *Fusarium* wilt and the susceptibility of the other local banana cultivar against *F. odoratissimum* (TR4).

## 2. Materials and Methods

### 2.1. Sample Collection

Symptomatic banana plants of various genome types were collected from 17 locations throughout Malaysia, including Sarawak's state. These samples were stored at the Biological Control Laboratory, Universiti Putra Malaysia, Selangor, Malaysia. Thirty-eight fungal isolates were obtained from symptomatic stems, corms, and roots (Table 1).

### 2.2. Morphological Identification of Fusarium Species

All isolates were grown on potato dextrose agar (PDA) medium for seven days in the incubator at 25 °C to observe the colony colour and growth rate.

Carnation Leaves Agar (CLA) was prepared with modification [17]. Carnation leaves were obtained from a local florist at Kea Farm Market, Cameron Highlands, Pahang. Carnation leaves were washed under running tap water for 1 h, cut into 1cm × 1cm, and dried at 45 °C for 2 h, where leaves should remain green but brittle. The dried leaves were sterilised by autoclaving at 121 °C for 15 min. Two pieces of sterilised carnation leaves were placed on 2% water agar before the agar solidified. A 5 mm mycelial plug was placed beside the leaves and incubated for 30 days at 12 h light and 12 h dark cycle at 25 °C to induce sporulation.

Spezieller Nährstoffarmer agar (SNA) were prepared according to Nirenberg [18]. The medium was autoclaved at 121 °C for 15 min and poured into the Petri dishes. A five mm mycelial plug was placed in the centre of the agar and incubated under continuous light at 25 °C for 10 days to promote branching of the conidiophore. Aerial conidiophores will be observed for microconidia and sporodochia production.

After 7, 10, and 30 days of incubation in CLA, SNA, and PDA, microscopic observation was conducted by preparing the fungal isolates with lactophenol cotton blue (LCB) dye

and Shear Mounting Fluid [19]. Microscopic images were taken at 400× and 1000× magnification using Canon EOS D1000, and images were analysed using ImageJ 1.52q [20]. Size measurement of 30 conidia per representative group was recorded.

**Table 1.** Sample collection locations, parts, isolate numbers, and GenBank accessions.

| Sample Location | GPS Coordinate | Parts Sampled | Sample Code | GenBank Accessions | |
|---|---|---|---|---|---|
| | | | | TEF-1a | HIS H3 |
| Universiti Putra Malaysia, Selangor | N 2 59.510 E 101 42.963 | Stem | MUCC2830 | LC545806 | LC545766 |
| Kluang, Johor | N 2 02.996 E 103 17.847 | Stem | MUCC2831 | LC545832 | LC545774 |
| Batu Pahat, Johor | N 1 50.863 E 103 05.339 | Root | MUCC2832 | LC545800 | LC545770 |
| Batu Pahat, Johor | N 1 52.001 E 102 55.593 | Stem | MUCC2833 | LC545831 | - |
| Ayer Hitam, Johor | N 1 56.237 E 103 11.050 | Stem | MUCC2834 | LC545830 | LC545775 |
| Ayer Hitam, Johor | N 1 56.237 E 103 11.050 | Stem | MUCC2835 | LC545829 | LC545776 |
| Ayer Hitam, Johor | N 1 56.237 E 103 11.050 | Stem | MUCC2836 | LC545833 | LC545777 |
| Jasin, Melaka | N 2 18.912 E 102 25.800 | Soil | MUCC2837 | LC545828 | LC545778 |
| Kuala Pilah Tengah, Negeri Sembilan | N 2 41.494 E 102 11.839 | Corm | MUCC2838 | LC545827 | LC545779 |
| Jabatan Pertanian Lekir, Perak | N 4 08.673 E 100 43.742 | Corm | MUCC2839 | LC545836 | - |
| Kuala Pilah Tengah, Negeri Sembilan | N 2 41.494 E 102 11.839 | Corm | MUCC2840 | LC545826 | LC545780 |
| Kuala Pilah Tengah, Negeri Sembilan | N 2 41.494 E 102 11.839 | Stem | MUCC2841 | LC545825 | LC545781 |
| Kuala Pilah Tengah, Negeri Sembilan | N 2 41.494 E 102 11.839 | Stem | MUCC2842 | LC545824 | LC545782 |
| Universiti Putra Malaysia, Selangor | N 2 59.510 E 101 42.963 | Stem | MUCC2843 | LC545823 | - |
| Universiti Putra Malaysia, Selangor | N 2 59.510 E 101 42.963 | Stem | MUCC2844 | LC545822 | LC545783 |
| Universiti Putra Malaysia, Selangor | N 2 59.510 E 101 42.963 | Stem | MUCC2845 | LC545821 | LC545784 |
| Universiti Putra Malaysia, Selangor | N 2 59.510 E 101 42.963 | Stem | MUCC2846 | LC545805 | - |
| Kampung Poh, Bidor, Perak | N 4 05.185 E 101 20.019 | Stem | MUCC2847 | LC545820 | LC545785 |
| Kampung Banir, Bidor, Perak | N 4 12.849 E 101 10.580 | Stem | MUCC2848 | LC545819 | LC545786 |
| Kampung Poh, Bidor Perak | N 4 05.185 E 101 20.019 | Root | MUCC2849 | LC545818 | LC545787 |
| Chetok, Kelantan | N 6 02.727 E 100 12.341 | Stem | MUCC2850 | LC545835 | - |
| Chetok, Kelantan | N 6 02.727 E 100 12.341 | Soil | MUCC2851 | LC545834 | - |
| Kampung Perlis, Pulau Pinang | N 5 18.796 E 100 12.341 | Root | MUCC2852 | LC545817 | LC545788 |
| Kampung Perlis, Pulau Pinang | N 5 18.796 E 100 12.341 | Root | MUCC2853 | LC545816 | LC545789 |
| Kampung Karu, Sarawak | N 1 17.119 E 110 16.936 | Stem | MUCC2854 | LC545804 | LC545768 |
| Kampung Pulau Manis, Terengganu | N 5 14.763 E 103 01.730 | Corm | MUCC2855 | LC545799 | LC545771 |
| Kampung Sungai Maong, Sarawak | N 1 32.537 E 110 18.313 | Corm | MUCC2856 | LC545815 | LC545790 |
| Kampung Sungai Maong, Sarawak | N 1 32.537 E 110 18.313 | Corm | MUCC2857 | LC545814 | LC545791 |
| Kampung Karu, Sarawak | N 1 17.119 E 110 16.936 | Corm | MUCC2858 | LC545803 | LC545796 |
| Chetok, Kelantan | N 6 02.727 E 102 08.661 | Corm | MUCC2859 | LC545813 | LC545792 |
| Tumpat, Kelantan | N 6 07.184 E 102 13.275 | Corm | MUCC2860 | LC545812 | LC545793 |
| Tumpat, Kelantan | N 6 07.184 E 102 13.275 | Corm | MUCC2861 | LC545811 | LC545794 |
| Sungai Atong, Pahang | N 3 52.953 E 103 09.777 | Stem | MUCC2862 | LC545810 | LC545798 |
| Kampung Ria Semantan, Pahang | N 3 56.334 E 101 50.423 | Corm | MUCC2863 | LC545809 | LC545795 |
| Kampung Tanjung Besar, Pahang | N 3 28.015 E 102 28.121 | Root | MUCC2864 | LC545808 | LC545796 |
| Kampung Ria Semantan, Pahang | N 3 56.334 E 101 50.342 | Root | MUCC2865 | LC545807 | LC545797 |
| Kampung Ria Semantan, Pahang | N 3 56.334 E 101 50.342 | Corm | MUCC2866 | LC545802 | LC545772 |
| Kampung Ria Semantan, Pahang | N 3 56.334 E 101 50.342 | Corm | MUCC2867 | LC545801 | LC545773 |

### 2.3. Molecular Identification of Fusarium Species

To identify the *Fusarium* species related to *Fusarium* wilt, two protein-coding regions known as species-level barcodes, histone H3 coding gene (*H3*) and translation elongation factor 1-alpha coding gene (*TEF-1α*), were analysed.

All the isolates of *Fusarium* spp. were grown on PDA for seven days. Genomic DNA was extracted from isolates using the cetyl trimethyl ammonium bromide (CTAB) method described by Umesha et al. [21].

The *H3* gene was amplified using the H3-1a and H3-1b primers [12], and the TEF-1a gene was amplified using primer EF1 and EF2 [16], respectively. To confirm the presence of *F. odoratissimum*, all fungal DNA extract was subjected to Foc-TR4 specific PCR amplification analysis using Foc-TR4 specific primers, FocTR4-F and FocTR4-R [22].

The PCR condition was as follows: for *H3* gene [12]: PCR was carried out in a 12.5 μL reaction volume containing 8.2 μL ddH$_2$O, 0.38 μL MgCl$_2$ (Nippon Gene), 1.25 μL buffer solution (Nippon Gene), 1 μL dNTPs (Nippon Gene), 0.31 μL of forward primer H3-1a, and 0.31 μL of reverse primer H3-1b and 1 μL DNA template. PCR amplification was performed as following protocols; initial denaturation 94 °C for 2 min; 30 cycles of denaturation at 94 °C for 1 min, annealing at 56 °C for 1 min and elongation at 72 °C for 1 min; final extension of 72 °C for 5 min; for *TEF-1α* [16]: PCR was carried out in a 12.5 μL reaction volume containing 6.25 μL master mix, 8.2 μL ddH$_2$O, 0.38 μL MgCl$_2$ (Nippon Gene), 1.25 μL buffer solution (Nippon Gene), 1 μL dNTPs (Nippon Gene), 0.31 μL of forward primer EF1, and 0.31 μL of reverse primer EF2 and 1 μL DNA template. PCR amplification was performed as following protocols; initial denaturation 95 °C for 2 min; 35 cycles of denaturation at 95 °C for 30 s, annealing at 57 °C for 30 s and elongation at 72 °C for 1 min 30 s; final extension of 72 °C for 10 min; and for TR4 specific primer [22]: PCR was carried out in a 12.5 μL reaction volume containing 6.25 μL master mix, 4.65 μL ddH$_2$O, 0.3 μL of forward primer FocTR4-F, and 0.3 μL of reverse primer FocTR4-R and 1 μL DNA template. The PCR amplification was performed as the following protocols; initial denaturation at 95 °C for 2 min; 30 cycles of denaturation at 95 °C for 1 min, annealing at 60 °C for 1 min and elongation at 72 °C for 3 min; final extension of 72 °C for 10 min.

The sequences of all purified PCR products of *H3* and *TEF-1α* genes were analysed on an Applied Biosystems 3730xl DNA Analyzer (Life Technologies, Carlsbad, CA, USA) installed at the Advanced Science Research Promotion Center, Mie University, Mie, Japan. Sequences were assembled and manually edited using MEGA 7 version software [23]. All sequences were later re-assembled and aligned with similar sequences retrieved from the GenBank database (Supplementary Material).

### 2.4. Phylogenetic Analysis

Phylogenetic analysis in this study was based on Bayesian inference (BI), maximum-likelihood (ML) and maximum parsimony (MP). The MP analysis was conducted by PAUP v.4.0 b8 [24], where heuristic search options with 1000 random taxon addition and TBR were used as branch swapping algorithms were applied. ML analysis was conducted with raxmlHPC-PTHREADS [25], and branch strength was tested by bootstrap analysis by 1000 replication. Bayesian inference (BI) analysis was performed by using BEAST v.2.5.1 [26]. The Markov Chain Monte Carlo (MCMC) algorithm was used to calculate the posterior probability (PP) whereby the settings were adjusted to run for 10,000,000 generations and sampled at every 1000 generations. The initial 25% of phylogenies were discarded as the "burn-in" phase, and posterior probability was determined from the consensus phylogenies.

### 2.5. Pathogenicity Test

Pathogenicity test was conducted over 30 days in triplicate by using an isolate that was confirmed to be of high virulence [10] on five different cultivars of local banana, consisting of banana plantlet of genome type AAA (Dwarf Cavendish), AA (Lakatan), AAB (Raja and Laknau), and ABB (Saba). The inoculum was the conidial suspension of *F. odoratissimum* (MUCC2841), which was prepared by mung bean medium (MBM: 5 g mung bean with 1 L of ddH$_2$O) [27]. A seven day old mycelial plug was inoculated to MBM. At seven days of inoculation on a 100 rpm rotary shaker at 25 °C, the medium was filtered using two layers of sterile cheesecloth to remove the hyphal fragment. The conidia concentration was counted using a hemocytometer, with $10^6$ conidia per mL. Finally, 200 mL of the inoculum was directly added to a close potting system of the banana plantlet.

The severity of wilting was rated following the disease scale, where 1 indicates no symptom/healthy, 2 denotes initial yellowing mainly on the lower leaves, 3 means yellowing of all the lower leaves, including some discolouration on the younger leaves, 4 indicates intense yellowing on all leaves, and 5 means "plant dead" or complete wilting [10]. Additionally, pseudostem splitting was observed, and the plantlet was uprooted, cleaned,

and cut at rhizome, observing for corm rot, and rated as the following: 1 indicates no discolouration observed, 2 means discolouration on isolated points, 3 denotes about 30% discolouration, 4 indicates up to 50% discolouration, 5 means discolouration up to 90%, and 6 denotes plant decay [27].

Disease severity percentage was calculated by using the formula,

$$DS, \% = \frac{\Sigma(a \times b)}{N \cdot c} \times 100\%, \tag{1}$$

where $\Sigma$ ($a \times b$): sum of symptomatic plant and their score scale, $N$: total number of sampled plants, and $c$: highest score scale.

### 2.6. Statistical Analysis

One way analysis of variance (ANOVA) was performed to measure the conidia size and the disease severity of inoculated plantlets for the pathogenicity test. In addition, Fisher's least significant difference (LSD) was performed to determine the significant differences between groups, where the result was considered significantly different at a 95% confidence level. The statistical analysis was conducted by using Statistical Analysis System (SAS) University Edition software [28].

### 3. Results

### 3.1. Morphological Identification

After seven days of incubation on the PDA, the examined 38 isolates could be clustered into four different groups based on the cultural characters. Colony characteristics and growth rate of the isolates are listed as follows (Table 2). For each group, a representative isolate was chosen randomly for further analysis, which were MUCC2839 (Group 1), MUCC2841 (Group 2), MUCC2867 (Group 3), and MUCC2858 (Group 4).

**Table 2.** Characteristic culture morphology and growth rate of isolates grouping.

| Group | Isolates | Culture Characteristic | | | Growth Rate (PDA) cm/day |
|---|---|---|---|---|---|
| | | PDA | SNA | CLA | |
| 1 | MUCC2839 | Magenta pigmentation on both sides of the plate | Sparse mycelia | "Wet" mycelia with pale yellow to orange sporodochia | 0.57 ± 0.20 |
| 2 | MUCC2841 | White colony with cottony and dense mycelia | Floccose mycelia | Sparse mycelia, pale yellow to white sporodochia | 0.60 ± 0.10 |
| 3 | MUCC2867 | White colony with cottony and floccose mycelia | Abundance mycelia | Dense mycelia with yellow sporodochia | 0.48 ± 0.13 |
| 4 | MUCC2858 | White colony with purple pigmentation, floccose and sparse mycelia | Abundance floccose mycelia | Sparse mycelia with yellow to orange sporodochia | 0.52 ± 0.11 |

PDA: Potato Dextrose Agar; CLA: Carnation Leaves Agar; SNA: Spezieller Nährstoffarmer Agar.

Representative strains showed typical colonies of *Fusarium* spp. (Figure 1; Table 2). The morphological characteristics of the fungal pathogen on the CLA medium showed no significant differences between the macroconidia and microconidia sizes and shape of the isolates (Figure 2; Table 3).



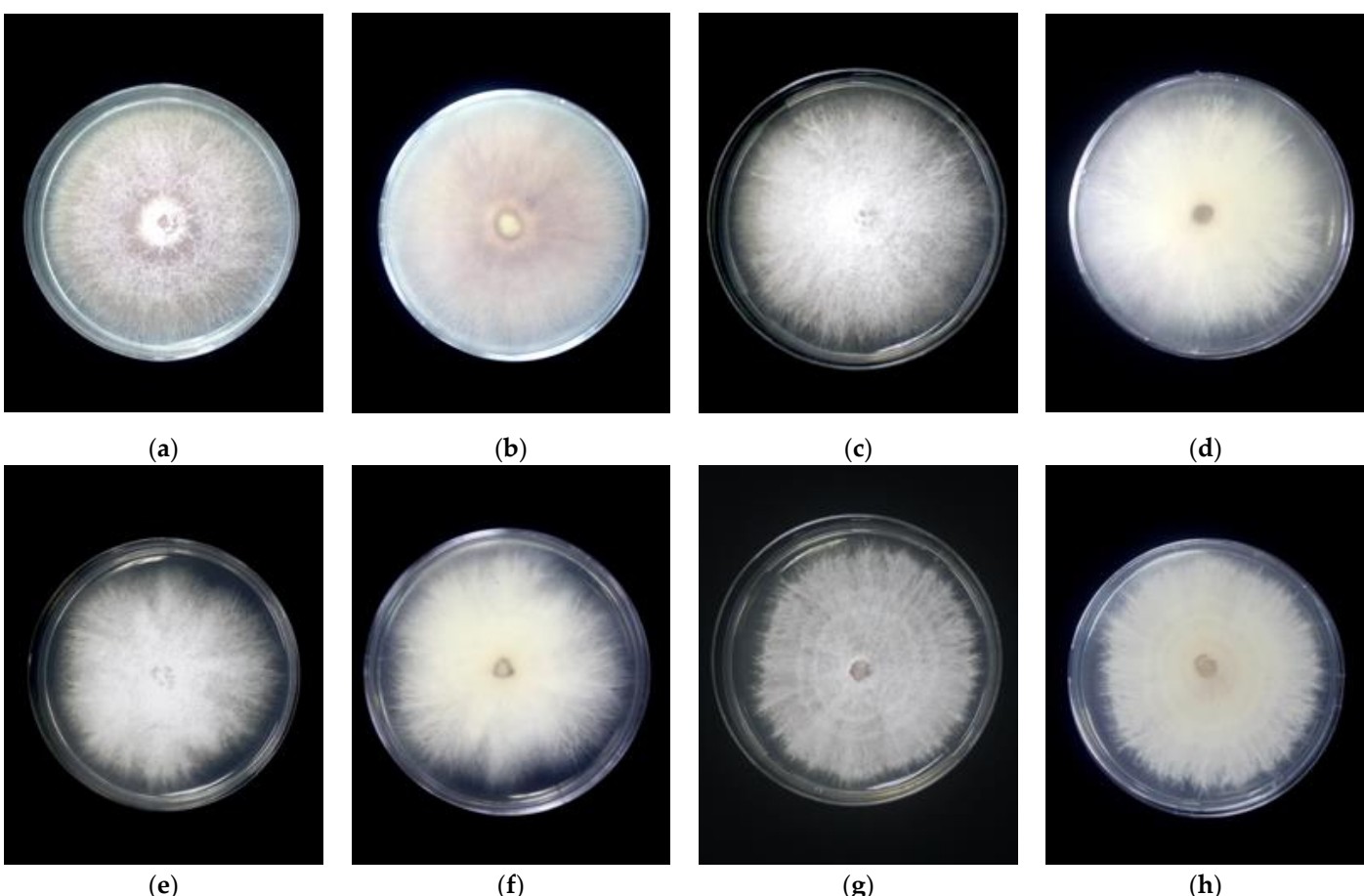

**Figure 1.** Isolates growth on PDA after seven days of incubation; (**a**,**b**) aerial and backplate view of the isolate MUCC2839; (**c**,**d**) aerial and backplate view of the isolate MUCC2841; (**e**,**f**) aerial and backplate view of the isolates MUCC2867; (**g**,**h**) aerial and backplate view of the isolates MUCC2858.

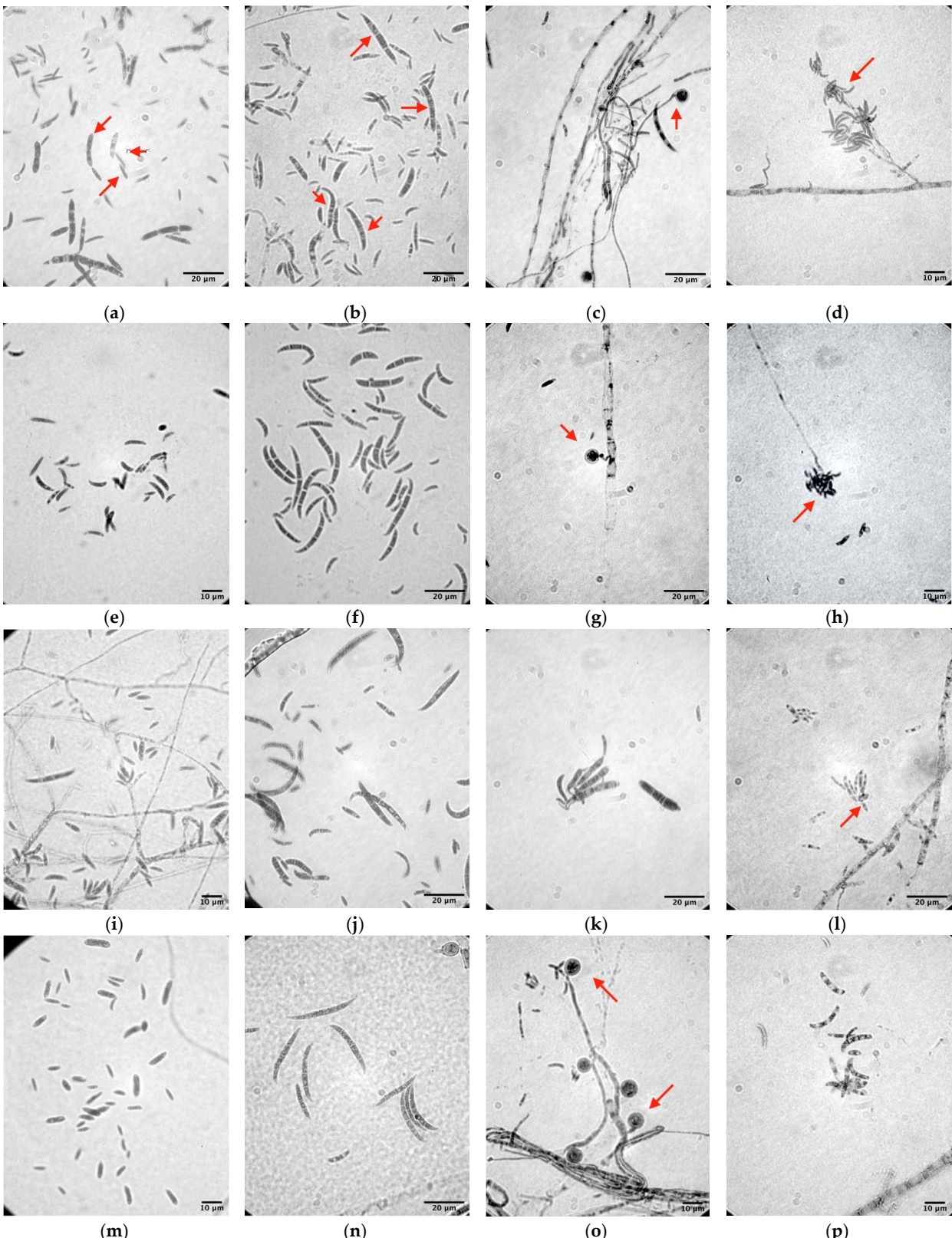

**Figure 2.** Micromorphology of MUCC2893: (**a**) microconidia; (**b**) macroconidia; (**c**) chlamydospores; (**d**) false head. Micromorphology of MUCC2841: (**e**) microconidia; (**f**) macroconidia; (**g**) chlamydospores; (**h**) False HEAD. Micromorphology of MUCC2867: (**i**) microconidia; (**j**) macroconidia; (**k,l**) false head. Micromorphology of MUCC2858: (**m**) microconidia; (**n**) macroconidia; (**o**) chlamydospores; (**p**) false head.

**Table 3.** Macro- and micro-conidia size, shape, and septation of representative isolates on CLA.

| Isolates | Macroconidia | | | | Microconidia | | | |
|---|---|---|---|---|---|---|---|---|
| | Length | Width | Septate | Shape | Length | Width | Septate | Shape |
| MUCC2839 | 15.07–30.99 (24.94 ± 4.09) | 1.39–3.94 (2.59 ± 0.53) | 3–5 | Falcate | 3.58–12.54 (7.02 ± 2.15) | 0.86–2.43 (1.45 ± 0.43) | 0–1 | Oval to ellipsoid |
| MUCC2841 | 16.89–33.84 (25.06 ± 4.94) | 1.76–3.65 (2.32 ± 0.54) | 0–6 | Falcate | 3.29–12.34 (6.97 ± 2.75) | 0.77–2.42 (1.48 ± 0.47) | 0–3 | Oval to ellipsoid |
| MUCC2867 | 19.93–31.85 (25.42 ± 3.65) | 1.03–3.94 (2.38 ± 0.65) | 0–5 | Falcate | 4.78–11.48 (6.13 ± 1.46) | 0.86–2.93 (1.64 ± 0.48) | 0–2 | Oval to ellipsoid |
| MUCC2858 | 16.97–38.07 (26.48 ± 5.41) | 1.25–3.27 (2.24 ± 0.61) | 0–4 | Falcate | 3.79–12.43 (7.14 ± 1.96) | 1.18–2.18 (1.64 ± 0.28) | 0–1 | Oval to ellipsoid |

*3.2. Molecular Analysis*

PCR amplification using H3 and TEF-1$\alpha$ genes was successful, and all sample sequences were deposited into the GenBank repository under the accession number LC545766 to 545836 (Table 1). All amplification by the H3 was successful. Nonetheless, a total of six samples were not successfully sequenced after several attempts. Reference strains were taken from two different databases for the H3 and TEF-1$\alpha$ genes. For the H3 gene, all reference sequences were taken from the Approved Strain Database of *Fusarium* Species from the National Agriculture and Food Research Organization, Japan. In contrast, for the TEF-1$\alpha$ gene, all reference sequences were retrieved from the GenBank database. On generated trees using the H3 gene sequences matrix (Figure 3), 33 examined isolates formed a well-supported clade of *Fusarium oxysporum* species complex with two isolates of *F. oxysporum* (MAFF 410171, MAFF 410172) (MP-BS/ML-BS/Bayesian PP = 99/100/1).

In the analysis by *TEF-1$\alpha$* coding gene sequences, known as a species barcode region for *Fusarium fujikuroi* species complex, the generated tree (Figure 4) revealed the phylogenetic position of examined isolates. Three isolates from the phylogenetic tree were recognised as *Fusarium sacchari* (MP-BS/MP-BS/Bayesian PP = 100/100/1). Meanwhile, 27 isolates were clustered with *F. odoratissimum* (MP-BS/MP-BS/Bayesian PP = 90/68/-). In addition, although the low statistical support, two isolates were recognised as *Fusarium grosmichelii*, four isolates were identified as *Fusarium oxysporum* s. lat., and two isolates as *Fusarium* spp. The summary of the findings was presented in Table 4.

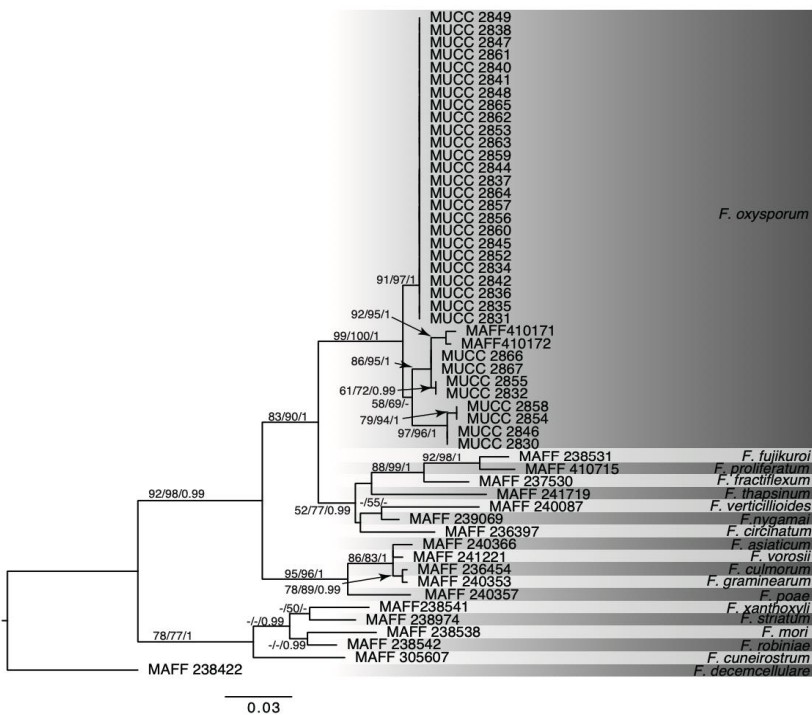

**Figure 3.** Phylogenetic tree constructed using a maximum-likelihood method based on the *H3* gene sequence. The MP and ML bootstrap values and Bayesian posterior probability (PP) value are denoted near the branch (MP/ML/PP) where MP/MP/PP (>50/50/0.96) indication of support.

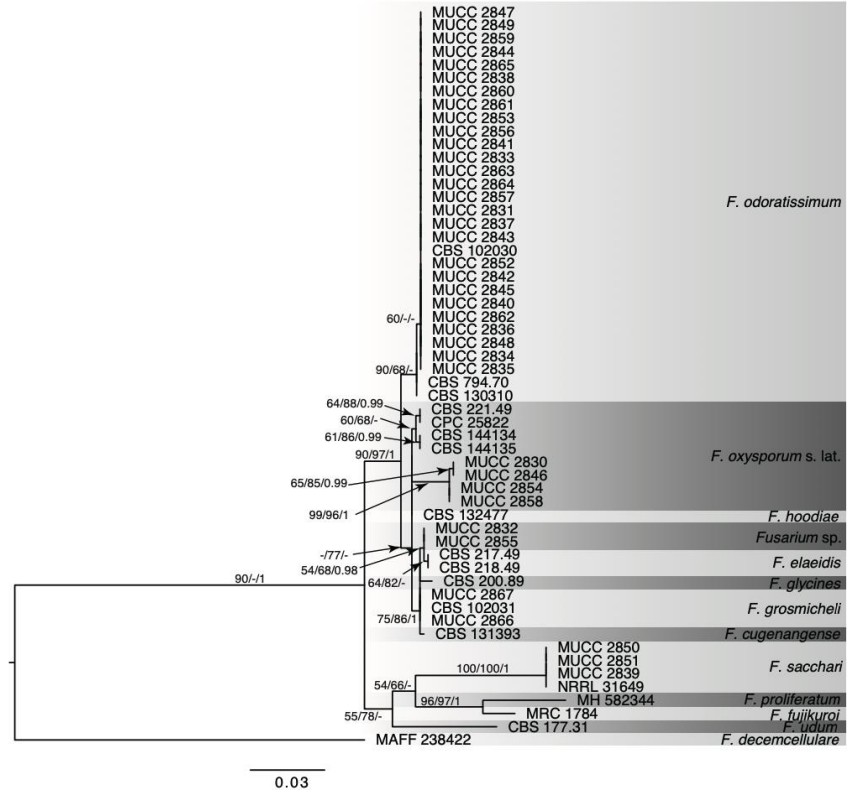

**Figure 4.** Phylogenetic tree constructed using a Maximum-likelihood method based on the *TEF-1α* gene sequence. The MP and ML bootstrap values and Bayesian posterior probability (PP) value are denoted near the branch (MP/ML/PP) where MP/MP/PP (>50/50/0.96) indication of support.

**Table 4.** Summary of the findings for all isolates obtained in this study using the *H3* and *TEF-1α* genes amplification.

| Sample Code | H3 | TEF-1α |
|---|---|---|
| MUCC2830 | *F. oxysporum* | *F. oxysporum s. lat.* |
| MUCC2831 | *F. oxysporum* | *F. odoratissimum* |
| MUCC2832 | *F. oxysporum* | *F. oxysporum* |
| MUCC2833 | Not available | *F. odoratissimum* |
| MUCC2834 | *F. oxysporum* | *F. odoratissimum* |
| MUCC2835 | *F. oxysporum* | *F. odoratissimum* |
| MUCC2836 | *F. oxysporum* | *F. odoratissimum* |
| MUCC2837 | *F. oxysporum* | *F. odoratissimum* |
| MUCC2838 | *F. oxysporum* | *F. odoratissimum* |
| MUCC2839 | Not available | *F. odoratissimum* |
| MUCC2840 | *F. oxysporum* | *F. odoratissimum* |
| MUCC2841 | *F. oxysporum* | *F. odoratissimum* |
| MUCC2842 | *F. oxysporum* | *F. odoratissimum* |
| MUCC2843 | Not available | *F. odoratissimum* |
| MUCC2844 | *F. oxysporum* | *F. odoratissimum* |
| MUCC2845 | *F. oxysporum* | *F. odoratissimum* |
| MUCC2846 | Not available | *F. oxysporum s. lat.* |
| MUCC2847 | *F. oxysporum* | *F. odoratissimum* |
| MUCC2848 | *F. oxysporum* | *F. odoratissimum* |
| MUCC2849 | *F. oxysporum* | *F. odoratissimum* |
| MUCC2850 | Not available | *F. sacchari* |
| MUCC2851 | Not available | *F. sacchari* |
| MUCC2852 | *F. oxysporum* | *F. odoratissimum* |
| MUCC2853 | *F. oxysporum* | *F. odoratissimum* |
| MUCC2854 | *F. oxysporum* | *F. oxysporum s. lat.* |
| MUCC2855 | *F. oxysporum* | *F. oxysporum* |
| MUCC2856 | *F. oxysporum* | *F. odoratissimum* |
| MUCC2857 | *F. oxysporum* | *F. odoratissimum* |
| MUCC2858 | *F. oxysporum* | *F. oxysporum s. lat.* |
| MUCC2859 | *F. oxysporum* | *F. odoratissimum* |
| MUCC2860 | *F. oxysporum* | *F. odoratissimum* |
| MUCC2861 | *F. oxysporum* | *F. odoratissimum* |
| MUCC2862 | *F. oxysporum* | *F. odoratissimum* |
| MUCC2863 | *F. oxysporum* | *F. odoratissimum* |
| MUCC2864 | *F. oxysporum* | *F. odoratissimum* |
| MUCC2865 | *F. oxysporum* | *F. odoratissimum* |
| MUCC2866 | *F. oxysporum* | *Fusarium grosmichelii* |
| MUCC2867 | *F. oxysporum* | *Fusarium grosmichelii* |

The result of amplification using the Foc-TR4 specific primers shows that 27 isolates, grouped with *F. odoratissimum* reference strains on the *TEF-1α* phylogeny, were positively identified as *F. odoratissimum*.

*3.3. Pathogenicity Test*

All inoculated plantlets showed typical symptoms of *Fusarium* wilt, such as yellowing of leaves, chlorosis, and death of older leaves at 30 days post-inoculation (Table 5). In a more severe condition, plant wilting was observed (Figure 5b). In this study, all inoculated plantlets showed foliar symptoms of infection with a different degree of severity. Apart from the foliar symptom, the banana plant was also observed lodging of plantlet, splitting at pseudostem, and corm rotting, the typical symptoms of *Fusarium* wilt. All five examined cultivars of inoculated bananas showed the splitting symptom at the pseudostem (Figure 5b,e,h,k,n). Although the degree of severity of the foliar symptom varied at the different cultivars, the internal symptoms of all cultivars showed severe corm rotting and discolouration (Figure 5c,f,i,l,o).

**Table 5.** Evaluation of susceptibility of local varieties of banana against *Fusarium odoratissimum*.

| Banana Cultivar (Genome Type) | Pathogenicity Rating Leaf [a] | Corm [b] | Notes |
|---|---|---|---|
| *Musa acuminata cv.* 'Dwarf Cavendish' (AAA) | 3 | 5 | Discolouration of younger leaves; pseudostem splitting; corm rot and discolouration. |
| *Musa acuminata* cv. 'Lakatan' (AA) | 4 | 6 | All leaves yellowing; pseudostem splitting; corm rotted and discoloured. |
| *Musa × paradisiaca* cv. 'Raja' (AAB) | 1 | 5 | Initial yellowing on older leaves; pseudostem splitting; corm rot and discolouration. |
| *Musa × paradisiaca* cv. 'Laknau' (AAB) | 2 | 5 | Older leaves yellowing; pseudostem splitting; corm rot and discolouration. |
| *Musa × paradisiaca* cv. Saba (ABB) | 2 | 5 | Older leaves yellowing; pseudostem splitting; corm rot and discolouration. |

[a] Severity of wilting was rated following the disease scale [10]; 1: no symptom/healthy, 2: initial yellowing mainly on the lower leaves, 3: yellowing of all the lower leaves including some discolouration on the younger leaves, 4: intense yellowing on all leaves, 5: plant dead/complete wilting. [b] Corm rot severity was rated following the disease scale [27]: 1 indicates no discolouration observed, 2 means discolouration on isolated points, 3 denotes about 30% of discolouration, 4 indicates up to 50% discolouration, 5 means discolouration up to 90%, and 6 denotes corm decay.

Statistical analysis shows a significant difference between control plantlets and inoculated plantlets for both foliar and corm symptoms as presented in Table 6. To verify the symptoms were caused by the inoculum, the pathogen was re-isolated from the corm of inoculated plantlets. It was identified as *F. odoratissimum* based on the morphology and PCR using the TR4 specific primer, which fulfils Koch's postulates.

**Table 6.** Leaf and corm discolouration percentage of control and inoculated plantlet.

| Banana Cultivar (Genome Type) | Leaf Control | Inoculated | Corm Control | Inoculated |
|---|---|---|---|---|
| *Musa acuminata cv.* 'Dwarf Cavendish' (AAA) | 26.38 [a] | 97.91 [d] | 0 [a] | 94.44 [c] |
| *Musa acuminata* cv. 'Lakatan' (AA) | 25.00 [a] | 70.07 [c] | 0 [a] | 88.89 [b] |
| *Musa × paradisiaca* cv. 'Raja' (AAB) | 25.00 [a] | 69.23 [c] | 0 [a] | 83.33 [b] |
| *Musa × paradisiaca* cv. 'Laknau' (AAB) | 25.00 [a] | 67.29 [bc] | 0 [a] | 83.33 [b] |
| *Musa × paradisiaca* cv. Saba (ABB) | 25.00 [a] | 59.23 [b] | 0 [a] | 83.33 [b] |

[a] no significant difference with control group. [b, c, d] significantly different from control group.

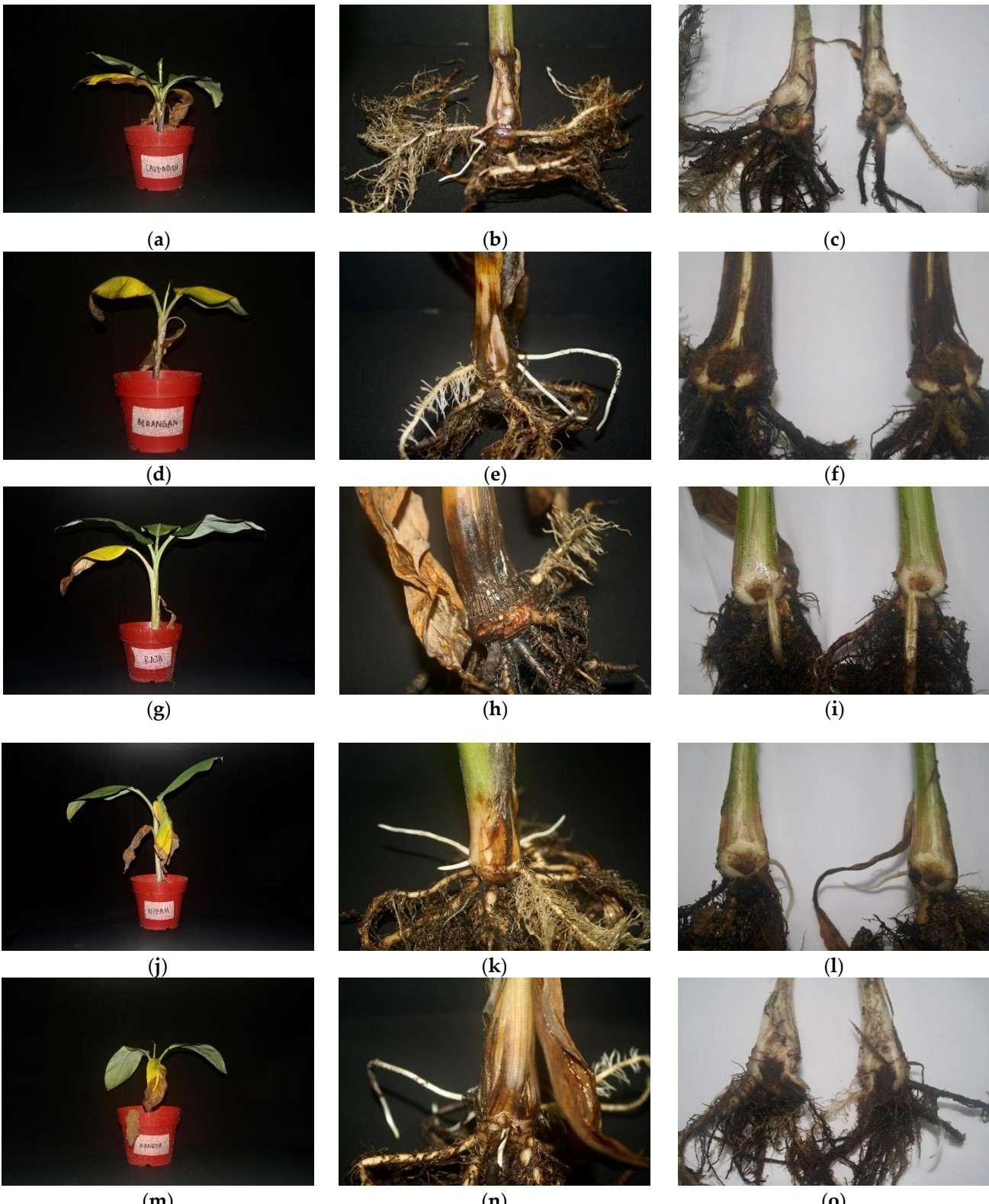

**Figure 5.** Pathogenicity test on local banana cultivars exhibiting wilting symptom (**a**) Dwarf Cavendish; (**d**) Lakatan; (**g**) Raja; (**j**) Laknau; (**m**) Saba. Plantlet shows pseudostem splitting on (**b**) Dwarf Cavendish; (**e**) Lakatan; (**h**) Raja; (**k**) Laknau; (**n**) Saba. Corm rotting (**c**) Dwarf Cavendish; (**f**) Lakatan; (**i**) Raja; (**l**) Laknau; (**o**) Saba.

## 4. Discussion

From the phenotypic characters, phylogenetic relationships, and Foc-TR4 specific PCR amplification analysis, these examined isolates were identified as *F. grosmichelii*, *F. odoratissimum*, *F. oxysporum* s. lat., and *F. sacchari*. There was an observable difference in colony morphology on the phenotypic characters of the isolates, which is the colony's colour as described by Leslie and Summerell [11]. Still, no significant difference was observed in conidia size. However, the mycelia of the genus *Fusarium* are floccose, sparse, or abundant with white to pale violet, and it mutates readily to flat and wet with yellow to orange. The size of macroconidia and microconidia was parallel with previous findings on the characteristics of *Fusarium* spp. The septation of both macro and microconidia was described by the *Fusarium* manual [11] and updated research on the morphology of *Fusarium* spp. [6].

For identification of *Fusarium* spp., the key features are cultural and morphological characteristics on the artificial media, SNA [18], CLA [17], and PDA. However, these characteristics could not distinctly differentiate at the *forma specialis* or the race of *F. oxysporum* species complex. Therefore, a more detailed examination, such as molecular characterization, is required to identify the species to grasp the fungal biodiversity and true pathogen related to the wilt of bananas, especially in the developing countries in which the plantations are developed.

In this study, we used the *H3* and *TEF-1α* coding genes for phylogenetic analyses of species and the species complex of the *Fusarium* genus [6,12,16]. Zeng et al. [14] insisted that the *H3* and *TEF-1α* coding genes have similarly good resolution on the internal groups within Foc isolates, related to the variety of pathogenicity. Our results showed that isolates of *F. odoratissimum*, identified by the specific primer analysis, formed a monophyletic clade on the *H3* and *TEF-1α* phylogenetic trees despite the differentiation of the pathogenicity. The identification using both the *H3* and *TEF-1α* coding gene sequences could only be conducted up to the genus level for the genus *Fusarium*. In this study, the results revealed that the *TEF-1α* gene was better in the detection of *F. odoratissimum* (Foc-TR4), known as the primary pathogen of *Fusarium* wilt for various genomes types of bananas. The result produced a more specific finding from the *TEF-1α* analysis as this could be due to the high volume of research conducted on the identification of *Fusarium* spp. Thus, this gene yielded more datasets for comparison [6,9,10,16,27,29]. One possible reason for the better result obtained using *TEF-1α* is that this protein is highly conservative and was identified as an essential genetic marker in the *Fusarium* study [12,30]. The presence of SNP in the TEF region was exploited to develop a specific primer for the detection of *F. odoratissimum* [22].

The diversity in the *Fusarium* spp. could be founded, despite the lack of sexual reproduction due to horizontal gene transfer [6,31], resulting in the formation of new lineages. However, as for *F. odoratissimum*, they usually were of the clonal population instead of multiple origins [10]. These results suggest that *F. odoratissimum* in this study was of the same lineage. On the other hand, *Fusarium* spp. related to the wilted banana has been reported in the 1990s in Malaysia, there are several hypotheses to the origin of these *Fusarium* species in the Indo-Malaya region [4] and that fungal pathogen co-evolved with the host plant itself. From the previous studies of diversity of strains based on the TEF-1α coding gene [9,10,32], numerous strains of *Fusarium oxysporum* f. sp. *cubense*, of which the race is unknown according to Leong et al. [9] or Tropical Race 4 Lineage V and Lineage I-II-V in Wong et al. [10], are widely disseminated across Peninsular Malaysia and were found from the different local banana cultivars. However, in this study, only *F. odoratissimum* (Foc-TR4), which is monophyletic on *TEF-1α* and *H3* phylogenetic trees, was detected from the entire of Malaysia. These results may indicate the simplification of the diversity of *F. odoratissimum* by applying selection pressure of agricultural chemicals or environmental events such as simplifying the variety of cultivating bananas.

*Fusarium sacchari* isolated from symptomatic banana plants is a member of *Fusarium fujikuroi* species complex. According to a recent taxonomical study by Maryani et al. [33], it is non-pathogenic to Cavendish and spends an endophytic life cycle.

However, there is a probability that the banana plant serves as an intermediary host for Pokkah boeng disease of sugarcane [34]. Although the total crop yields are not significant in Malaysia, *F. sacchari* as a sugarcane pathogen should be observed in the field.

In our study, two isolates of *Fusarium grosmichelii*, which is *F. oxysporum* f. sp. *cubense* race 1, which is known to affect cultivar 'Gros-Michel', 'Silk', and 'Pome' in Malaysia, were obtained. These isolates were obtained from Pisang Nangka (AAB). A previous study showed that the banana of genome AAB is susceptible to *F. grosmichelii* (R1) [1,2,35]. Thus, this race is the causal pathogen of the first wave of *Fusarium* wilt of banana that wiped out the world's cultivar 'Gros Michel'. Although the diversification of cropping varieties with local varieties is a generally efficacious approach for controlling the diseases, this result shows that local varieties are still facing the re-emergence of the forgotten disease.

For the pathogenicity test, the high virulence isolate of *F. odoratissimum* (MUCC2841) was inoculated to the local banana varieties in the previous study [10]. The foliar symptoms of all inoculated plants indicated susceptibility to *F. odoratissimum*. The foliar symptoms, including wilting and chlorosis of leaves, vary for each banana cultivar, even though it was infected with the same fungal isolate (Table 5). Cultivar 'Lakatan' (AAA) suffered the most severe infection as seen on the foliar symptom, whereby all of the leaves that were yellowing started to wilt. For pseudostem and internal symptoms, all inoculated plantlets showed pseudostem splitting, corm rot, and internal discolouration.

Although foliar symptoms suggest the establishment of infection by a pathogen, it is often an unreliable criterion for disease severity assessment [36]. García-Bastidas et al. [27] stated that morphological changes observed on foliar symptoms might be due to the trimming of the root for the root dip method. It causes atypical chlorosis resulting in similar symptoms of *Fusarium* wilt [31]. Furthermore, Chen et al. indicated that the wilting symptom without corm discolouration was uninfected at the xylem and was considered uninfected by the fungal pathogen [34,36]. The internal symptoms on pseudostem and corm are a more reliable parameter for disease assessment for *Fusarium* wilt from the previous study, which applied a different inoculation method yield a consistent result in the discolouration [36].

In the detection of *Fusarium* wilt, the molecular method was developed for the rapid detection of samples collected in the field [23]. Due to the lack of accessibility to access the international collection of VCG samples and the longer time needed to conduct VCG testing, the TR4 specific method is a highly specific detection method with high accuracy [27]. A previous study conducted on Malaysian isolates by Leong et al. [9] concluded that Malaysian *Fusarium oxysporum* is polyphyletic. However, there is no mention of its pathogenicity or race. A study conducted by Wong et al. [10] shows that all *F. odoratissimum* (TR4) are monophyletic. This study agrees with both findings, whereby variations could be founded in the *F. oxysporum* species complex, but *F. odoratissimum* (TR4) itself is monophyletic. A previous study conducted by Dita et al. [27] also mentions that the VCG of the *Fusarium* itself does not correlate with pathogenicity.

Our results support the previous study that *M. acuminata* (AAA) variants, cultivar 'Berangan' (AAA), are highly susceptible to *F. odoratissimum* [10]. The banana cultivars used in this study are in extremely high demand and widely planted in Malaysia. Once this fungus is transmitted on the banana plantation, it may lead to severe economic damage. Although diversification of different crop varieties is generally a productive approach to control disease, the *F. odoratissimum* was found to infect all banana cultivars in this study regardless of genotype. Therefore, resistance cultivar is one of the best choices for managing *Fusarium* wilt [7]. However, it is not obtained yet for cropping in a standard field except the transgenic Cavendish bananas in the test fields [37]. Therefore, early detection with rapid diagnoses and digging up the diseased plant and soil disinfection are crucial for controlling this disease as it stands now. Further studies with local varieties of bananas are required to understand the diversity of tolerance against the various strains of *Fusarium* wilt pathogens.

## 5. Conclusions

The findings of this study demonstrate that there is a diverse community of *Fusarium* spp. incorporated to *Fusarium* wilt disease of banana in Malaysia. Susceptibility of local commercial banana cultivar against Fusarium wilt was also assessed at a preliminary level, regardless of the genotype of the banana plants. Thus, this study provides new insight into the diversity of *Fusarium* spp. presence in the Malaysian banana plantation. As there is no practical method to control this disease, it would be challenging for the local banana breeders to search for resistance cultivars for cropping purposes.

**Supplementary Materials:** The following are available online at https://www.mdpi.com/article/10.3390/agronomy11101955/s1, Figure S1: His H3 Bayesian tree with posterior probability (PP) with 1,000,000 generation. MCMC chain were sampled at every 1000 generation. Branch length unit is substitution per site. Figure S2: His H3 Maximum parsimony (MP) tree with bootstrap value. Tree length (TL) = 343, Consistency Index (CI) = 0.633, Retention Index (RI) = 0.833, Rescaled Consistency (RC) = 0.527. Figure S3: TEF-1a H3 Bayesian tree with posterior probability (PP) with 1,000,000 generation. MCMC chain were sampled at every 1,000 generation. Branch length unit is substitution per site. Figure S4: TEF-1a Maximum parsimony (MP) tree with bootstrap value. Tree length (TL) = 235, Consistency Index (CI) = 0.868, Retention Index (RI) = 0.929, Rescaled Consistency (RC) = 0.806. Table S1: GenBank accessions numbers of Fusarium reference strains for phylogenetic analyses.

**Author Contributions:** Conceptualisation, D.Z., G.V. and C.N.; methodology, A.H.U. and C.N.; software, Y.H.; validation, D.Z. and C.K.F.W.; formal analysis, A.H.U. and Y.H.; investigation, A.H.U.; resources, D.Z. and C.N.; data curation, A.H.U. and Y.H.; writing—original draft preparation, A.H.U.; writing—review and editing, D.Z., G.V., C.N. and C.K.F.W.; visualization, A.H.U. and Y.H.; supervision, D.Z.; project administration, D.Z.; funding acquisition, D.Z. and C.N. All authors have read and agreed to the published version of the manuscript.

**Funding:** This research was funded by the Ministry of Higher Education of Malaysia (MOHE) through Fundamental Research Grant Scheme (FRGS/1/2018/WAB01/UPM/02/9), GP-GERAN PUTRA/ 9551700 and GP-IPS/9546600. This project was partially supported by the Institute for Fermentation, Osaka (IFO), Japan, and JSPS KAKENHI (17K07837 and 20K06146 to CN).

**Institutional Review Board Statement:** Not applicable.

**Informed Consent Statement:** Not applicable.

**Acknowledgments:** The first author is grateful for the Tun Taib scholarship provided by the Sarawak Foundation, Malaysia. All of the authors would like to thank all involving party and colleague in involvement of this study.

**Conflicts of Interest:** The authors declare no conflict of interest.

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
