# Peer review of "Current Classification and Diversity of Fusarium Species Complex, the Causal Pathogen of Fusarium Wilt Disease of Banana in Malaysia"

_agronomy, doi:10.3390/agronomy11101955_

Round 1

Reviewer 1 Report

I can confirm that the subject matter of this study (Current Classification and Diversity of Fusarium Species Complex, the Causal Pathogen of Fusarium Wilt Disease of Banana in Malaysia) is interesting.

Comments to the Authors:

  • correct Conclusion - but the conclusion should not be a summary. Add a practical implications statement. In my opinion this section may be improved reducing data and giving few key message/take home message to the readers. An idea may be to synthetize in 3-5 bullet the key results of the study, evidences and recommendation. This improvement will increase clearness and readability. Add a practical implications statement

References Double check name spelling, year, page, etc. I will not check them; it is your responsibility (according to Instructions for Authors).

Author Response

Responding remarks by Reviewer 1

Thank you for the constructive comments and suggestions.

Comments to the Authors:

  1. Correct conclusion - but the conclusion should not be a summary. Add a practical implications statement. In my opinion this section may be improved reducing data and giving few key message/take home message to the readers. An idea may be to synthesize in 3-5 bullet the key results of the study, evidences and recommendation. This improvement will increase clearness and readability. Add a practical implications statement

The conclusion section was rewritten by reducing data, reflecting on the aim of this study, and adding in the implication of the study.

  1. References double-check name spelling, year, page, etc. I will not check them; it is your responsibility (according to Instructions for Authors).

The reference section was rechecked, and amendments were made where necessary.

Reviewer 2 Report

In the submitted manuscript by Dzarifah Zulperi and colleagues entitled “Current Classification and Diversity of Fusarium Species Complex, the Causal Pathogen of Fusarium Wilt Disease of Banana in Malaysia”, the authors examined thirty-eight isolates of Fusarium species for morphological characteristics on different media. Furthermore, they used the Histone gene (H3) and Translation Elongation Factor gene (TEF-1α) to construct phylogenetic trees of Fusarium isolates. Finally, pathogenicity test was conducted on local cultivars. The authors found that the phylogenetic trees of the genus Fusarium related to Fusarium wilt are in rich diversity and F. odoratissimum has pathogenicity to local banana cultivars.

This work provided some new insights on the Fusarium wilt in Malaysia. Overall, the manuscript is well-written, and generally, the data have a good presentation and statical analysis is clear. The aim and scope of Agronomy journal are in line with the current manuscript. However, there are some issues in the text that should be carefully addressed.

Major and minor comments,
1) In order to avoid phrase replication lines; 103-104 and 176-178 such as ‘One-way analysis of variance (ANOVA) on disease severity was performed, and Fisher Least Significant Difference (LSD) were used to determine significant difference between groups.’. You should provide a subsection 2.6 Statistical analysis.

2) Why do you not use the Chi-square test to find if the location or part sample has any participation or significance in Fusarium species identification? To my opinion, this test can lead you to extra results with significance to your approach.

Author Response

Responding remarks by Reviewer 2

Thank you for the thoughtful and constructive suggestions.

  1. In order to avoid phrase replication lines; 103-104 and 176-178 such as ‘One-way analysis of variance (ANOVA) on disease severity was performed, and Fisher Least Significant Difference (LSD) were used to determine significant difference between groups.’. You should provide a subsection 2.6 Statistical analysis.

Section 2.6 was added (Line 181-187) to avoid repetition, as suggested by the reviewer. 

  1. Why do you not use the Chi-square test to find if the location or part sample has any participation or significance in Fusarium species identification? To my opinion, this test can lead you to extra results with significance to your approach.

The sample size of our study is too small to use a Chi-square test for the significance of sampling location. However, our samples were collected all around Peninsular Malaysia and the Sarawak state to represent the general distribution of Fusarium infection across the country.
